# FINEKB: DOMAIN-ADAPTIVE ISSUE SUMMARIZATION AND CLUSTER-AWARE RETRIEVAL FOR SUPPORT KNOWLEDGE BASES

## ABSTRACT

Retrieving relevant knowledge base (KB) articles for enterprise support cases is difficult due to the semantic mismatch between noisy, verbose case descriptions and concise KB content. We present FineKB, a domain-adaptive issue–summarization and cluster-aware retrieval framework that addresses this gap through (i) a finetuned LLM trained on teacher-generated pseudo-summaries to normalize heterogeneous case narratives, (ii) per-KB multi-centroid clustering that models the diverse sub-problems associated with each KB article, and (iii) a confidence-adaptive hybrid inference mechanism that augments high-confidence vector search with selective content lookup and LLM reasoning for ambiguous cases. At inference time, raw case text is embedded and matched against this summary-structured index, avoiding runtime summarization while improving alignment. Experiments on large-scale enterprise data show that FineKB achieves 65.39% Recall@3, substantially outperforming KB-content dense retrieval (42.73%). To support reproducible research on noisy-to-structured retrieval, we release FineKB-Vectors, a vectorized dataset containing case-summary and KB-article embeddings.

## 1 INTRODUCTION

Enterprise IT support generates large volumes of unstructured troubleshooting artifacts—including customer communications, diagnostic logs, and engineer notes—recorded within a support ticket (Faheem et al., 2025; He et al., 2021). These artifacts constitute the operational narrative of a case but are often verbose, redundant, and noisy (Marcuzzo et al., 2022). As multiple engineers interact with a ticket, terminology drifts and irrelevant details accumulate, creating a substantial lexical and semantic mismatch between informal case descriptions and the concise prose of knowledge base (KB) solutions (Ko et al., 2006). This mismatch makes accurate retrieval uniquely challenging.

Conventional retrieval methods—keyword-based or dense—struggle in this setting because (i) the core issue is obscured by extraneous details, (ii) diverse narratives may correspond to the same KB solution, (iii) cases frequently depend on multiple KB articles for complete resolution rather than a single document, and (iv) domain terminology diverges from open-domain distributions (Zhao et al., 2024). These properties highlight the need for retrieval systems that can restructure case data and model the latent problem structure underlying KBs.

To address these challenges, we introduce FineKB, a domain-adaptive issue–summarization and cluster-aware retrieval framework. FineKB integrates three key components: (i) an LLM-driven normalization module that converts noisy, multi-authored case narratives into concise, domain-aligned issue summaries, (ii) a per-KB clustering strategy that constructs multi-centroid representations capturing the heterogeneous sub-problems historically associated with each KB article, and (iii) a confidence-adaptive hybrid inference mechanism that balances fast vector retrieval with selective lexical and generative reasoning for ambiguous queries. Together, these components form a unified approach that combines geometric similarity, structured modeling of case–KB relationships, and uncertainty-aware decision-making to bridge the semantic gap between raw support cases and curated KB documentation.

FineKB is built on three mutually reinforcing components:

**(1) LLM-driven pseudo-summarization for domain normalization.** A large teacher model generates concise, terminology-aligned issue summaries from raw case text. These summaries serve as weak supervision for training a compact student model (e.g., LLaMA-3.1-8B via QLoRA), producing consistent, domain-adapted representations that substantially reduce narrative noise.

**(2) Per-KB multi-centroid structural indexing with asymmetric retrieval.** Historical case summaries for each KB are embedded and clustered using hierarchical agglomerative clustering. The resulting multi-centroid structure models the heterogeneous sub-problems addressed by a KB article. During inference, raw case descriptions query the summary-based index asymmetrically, ensuring low-latency retrieval while leveraging the clustered semantic structure of the domain.

**(3) Confidence-adaptive hybrid inference.** A geometric confidence score, derived from the top-1 centroid similarity, governs a dynamic inference path. High-confidence queries enter a fast path featuring reliability-based re-ranking, while low-confidence queries activate a hybrid path that augments vector search with both KB-content retrieval and a single LLM-based KB article recommendation. This mechanism substantially improves accuracy while keeping LLM usage below 5%, maintaining practical throughput.

Experiments on a large-scale enterprise dataset demonstrate the effectiveness of this integrated approach. FineKB achieves 65.39% Recall@3 in the asymmetric configuration—outperforming dense retrieval over KB content by more than 20 points—and the hybrid inference further increases Recall@3 to 67.26% with only a marginal cost in throughput. We additionally show that QLoRA-trained 8B models match the retrieval accuracy of a 70B teacher, validating the efficiency of our domain adaptation pipeline.

Our contributions are summarized as follows:

- We present FineKB, a cluster-aware retrieval framework that integrates LLM-based normalization of noisy case narratives, per-KB multi-centroid representations capturing heterogeneous sub-problems, and a confidence-adaptive hybrid inference mechanism for robust, low-latency retrieval supporting real-world issue resolution workflows.

- We introduce an LLM-driven weak supervision pipeline in which a large teacher model generates concise, domain-aligned issue summaries used to fine-tune a compact student LLM, enabling high-quality representations without requiring human-labeled data.

- We release FineKB-Vectors, a dataset of anonymized case-summary and KB-article embeddings, supporting reproducible research on noisy-to-structured retrieval under enterprise distributions.

The remainder of the paper is structured as follows: Section 2 reviews related work; Section 3 describes the FineKB framework in detail; Section 4 presents empirical results; and Section 5 concludes with future directions.

## 2  RELATED WORK

Research on enterprise knowledge base (KB) retrieval, compressed text representations, and domain-adapted language models forms the foundation for FineKB. Traditional enterprise systems rely on keyword search, BM25 (Robertson & Zaragoza, 2009), or TF–IDF (Salton & Buckley, 1988), but these methods struggle with the substantial lexical and semantic mismatch between verbose, multi-authored case descriptions and the concise prose of KB articles. Dense retrievers such as BERT (Devlin et al., 2019), Sentence Transformers (Reimers & Gurevych, 2019), DPR (Karpukhin et al., 2020), and ColBERT (Khattab & Zaharia, 2020) improve semantic alignment but generally assume relatively clean or well-structured input. In enterprise IT support, case narratives frequently exhibit topic drift, inconsistent terminology, and irrelevant details, reducing the effectiveness of general-purpose dense retrieval.

To mitigate the semantic gap between queries and indexed documents, recent work has explored LLM-generated surrogate signals. HyDE (Gao et al., 2023) constructs hypothetical documents to guide dense retrieval; entity-aware summaries have been proposed for more reliable sponsored search (Liang et al., 2024); and multi-signal retrieval strategies have been studied in commercial sponsored search settings (Mohankumar et al., 2024). These approaches demonstrate the value of

generated intermediate representations, but they operate at the level of single proxy documents and do not model the underlying structural variability of many-to-many case–KB mappings. AutoKB (Sahay et al., 2025) addresses the problem from the document side by organizing KB content into hierarchical knowledge trees. FineKB complements these directions by structuring the *query space* itself through LLM-adapted summaries and per-KB clustering yielding multi-centroid KB representations, which explicitly capture heterogeneous sub-problems associated with each KB article.

Compressed text representations have long supported information retrieval. Extractive methods such as TextRank (Mihalcea & Tarau, 2004) and BERT-based extractors (Liu & Lapata, 2019) identify salient sentences, while abstractive models like T5 (Raffel et al., 2020) and BART (Lewis et al., 2020) generate compact semantic reformulations. Pseudo-document generation and query reformulation techniques (Lewis et al., 2021) further illustrate the effectiveness of intermediate surrogate representations for retrieval. However, enterprise support domains rarely contain human-written summaries, motivating weak supervision approaches such as pseudo-labeling (Lee, 2013). In this setting, evaluating generated summaries requires metrics that capture factual correctness and alignment with domain terminology; frameworks such as FactScore (Min et al., 2023) and Aragog (Eibich et al., 2024) provide LLM-based methods for assessing the factual consistency and relevance of generated text.

Domain adaptation of large language models (LLMs) is another closely related line of work. Techniques such as supervised fine-tuning (SFT) (Ouyang et al., 2022), LoRA (Hu et al., 2022), and QLoRA enable efficient adaptation of foundation models to specialized data distributions. These methods have proven effective in a range of domain-specific applications, including medical and customer-support tasks (Anisuzzaman et al., 2025; Yaoyang, 2025). Our work builds on this literature by using LLM-generated summaries as a weak supervision signal for QLoRA fine-tuning of a compact student model, yielding domain-specific issue representations suitable for scalable enterprise deployment.

Across these lines of research, most prior approaches focus on enriching a single query or document representation or impose structure only on the KB content. FineKB differs by jointly reorganizing both the query space and the KB space through domain-normalized LLM summaries and per-KB clustering that yields multiple centroids for each KB article. This multi-centroid representation explicitly models the heterogeneous issue variants historically associated with a single KB, supports asymmetric retrieval, and enables confidence-adaptive hybrid inference. Together, these components form a unified retrieval framework that directly addresses the noisy, many-to-many mapping problem in enterprise case–KB retrieval.

## 3 METHODOLOGY

FineKB introduces a domain-adaptive pipeline designed to bridge the semantic gap between unstructured support cases and structured documentation. The system is designed for **low latency** and **computational efficiency**, critical requirements for real-time enterprise support workflows. As illustrated in Figure 1, the architecture integrates large language models (LLMs) with cluster-aware indexing through three primary modules: the **Dataset Builder**, the **Issue Summarizer**, and the **Cluster-Aware Retrieval Engine**.

The system operates in two distinct phases:

- **Offline Phase (Training & Indexing):** Historical case logs are processed to generate pseudo-labeled summaries (via a teacher LLM) and construct KB-wise cluster centroids. This phase produces both the dense vector index and the training data used to fine-tune the efficient student model (SFT/QLoRA).

- **Online Phase (Inference):** When a new case arrives, the fine-tuned Issue Summarizer runs asynchronously to generate a concise issue summary for the agent. For retrieval, the *Retrieval Engine* queries the cluster index directly using the raw case text (the **Asymmetric Retrieval** configuration) to identify relevant KB articles in real time.

This split design ensures that computationally intensive tasks—such as clustering and teacher-model inference—are handled asynchronously, and crucially, that the most accurate retrieval path (raw case

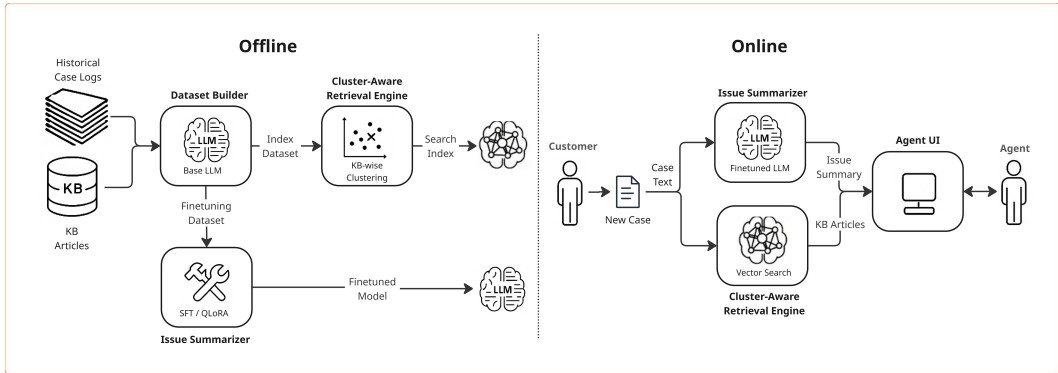

Figure 1: **FineKB architecture.** The system is organized into two phases. **Offline:** The Dataset Builder creates training targets and index structures from historical logs. **Online:** The fine-tuned Issue Summarizer and Cluster-Aware Retrieval Engine process new cases in real-time to surface summaries and relevant KB articles to the support agent.

text query) avoids the latency overhead of real-time summarization, enabling low-latency response times during active support workflows.

## 3.1 FineKB Dataset Builder

The FineKB Dataset Builder converts noisy historical case data into structured resources for *LLM fine-tuning* and *KB retrieval*. It performs three core functions: (i) consolidating case–KB associations, (ii) categorizing cases into meaningful case categories, and (iii) generating concise pseudo-summaries. Together, these steps establish a structured pipeline from raw logs to training-ready data and enable downstream FineKB modules.

**Case–KB Association.** Historical enterprise support logs contain free-form case descriptions, initially provided by customers and later expanded by support agents with troubleshooting details and linked KB articles. These agent-linked KBs serve as weak supervision signals, connecting cases to one or more relevant KBs. The Dataset Builder consolidates these many-to-many relationships into structured case–KB training pairs for model training. In this study, agent-linked KB associations are available; in domains lacking such labels, alternative weak supervision methods (e.g., heuristic matching, retrieval heuristics, or synthetic annotations) can be employed.

**Case Categorization.** Historical support logs often suffer from significant class imbalance, with common issues dominating while rare but critical failures are underrepresented. To create robust and representative datasets, we employ stratified sampling across issue types. In this study, the dataset is divided into 15 primary categories, as outlined in Table 3 (Appendix A). For cases lacking explicit issue-type labels, the case descriptions could be clustered to infer categories. Subsequent sampling across these clusters helps mitigate imbalance and ensures better coverage of diverse problem types.

**Pseudo-Summary Generation.** Enterprise support case descriptions are typically verbose, containing redundant or irrelevant information such as customer identifiers, troubleshooting notes, and metadata. Since human-annotated summaries are unavailable, we employ zero-shot issue summarization using a large language model (e.g., LLaMA-3.3-70B-Instruct). The model produces one-sentence pseudo-summaries that preserve domain-critical technical details—affected component, observed symptom, and exact error codes—while systematically filtering out non-technical information.

The generation process is guided by a structured prompt to enforce consistency and domain alignment:

> **Prompt Template**
>
> You are a support assistant for enterprise server products.
> Summarize the technical issue in one sentence.
> **Rules:**
> 1) Include product/model if available.
> 2) Specify affected component and symptom.
> 3) Retain exact error messages, codes, or event IDs.
> 4) Exclude resolutions, troubleshooting, customer identifiers, names, emails, phone numbers, addresses, dates, signatures, and case IDs.

For instance:

> **Raw Case Description.** "[Phone] System reports multiple DIMM errors during POST (Event ID: 0xA0). Customer attempted reseating. Contact: John D, 555-1234. Preferred Contact Method: Email"
>
> **Generated Pseudo-Summary.** "DIMM failure causing boot errors during POST (Event ID: 0xA0)."

This example highlights the model's ability to extract critical technical signals (component, symptom, error code) while discarding irrelevant details (customer metadata, troubleshooting actions). Pseudo-summaries are generated across stratified case categories to maintain balanced coverage of frequent and rare issue types, mitigating class imbalance. These outputs serve a dual role: (i) providing weak supervision for fine-tuning domain-adaptive LLMs, and (ii) acting as normalized intermediate representations that bridge noisy logs and structured KBs, thereby enhancing retrieval accuracy.

## 3.2 FineKB Issue Summarizer

The FineKB Issue Summarizer generates concise, domain-adaptive issue summaries from verbose case descriptions. Unlike generic summarization models, this module is explicitly adapted to enterprise terminology and trained to suppress irrelevant metadata while preserving critical technical signals such as affected components, symptoms, and exact error codes. The primary objective is to maximize the factual consistency and relevance of the summary, metrics which are later validated using LLM-as-a-Judge protocols (Section 4.2).

To achieve this, we fine-tune `LLaMA-3.1-8B` variants (Touvron et al., 2023)[1] using the pseudo-labeled summaries produced by the Dataset Builder. This process effectively **distills the teacher model's domain-specific summarization behavior** into the efficient student architecture. Two complementary adaptation strategies are employed:

- **Supervised Fine-Tuning (SFT):** updating model parameters with weakly supervised labels.

- **LoRA/QLoRA:** inserting low-rank adapters for efficient fine-tuning under limited computational resources.

Formally, given a raw case description $c_i$ and its pseudo-summary $\tilde{s}_i$, the Issue Summarizer $f_\theta$ is trained to minimize the cross-entropy loss:

$$\mathcal{L} = \sum_{i=1}^{N} \ell\big(f_\theta(c_i), \tilde{s}_i\big),$$

where $\ell$ denotes the token-level cross-entropy. This objective aligns the student model's outputs with the high-quality pseudo-labeled targets while adapting its vocabulary and style to enterprise-specific domains.

During both training and inference, we enforce a constrained prompting template to ensure consistency across generated summaries: outputs are restricted to a single sentence, explicitly mention

---

[1]Checkpoints obtained from HuggingFace: `https://huggingface.co/meta-llama`

affected components and symptoms, preserve error codes or event IDs verbatim, and exclude customer identifiers, troubleshooting steps, and other non-technical content.

By producing normalized and terminology-aligned representations, the FineKB Issue Summarizer provides support agents with concise, noise-free problem synopses that preserve key technical signals and reduce cognitive load during case investigation.

## 3.3 FineKB Cluster-Aware Retrieval Engine

The FineKB retrieval engine builds an index that captures the sub-problem diversity of KB articles by clustering historical case embeddings into multiple centroids per KB. This structure resolves a core limitation of standard dense retrieval, where collapsing all linked cases into a single vector obscures the heterogeneity of real-world issue descriptions.

We formulate the task as *retrieval* rather than *classification* because retrieval naturally supports (i) a dynamic and evolving KB catalog, (ii) generalization to unseen KBs, and (iii) many-to-many mappings where a single case may require multiple KB articles. In contrast, classification requires retraining whenever the KB repository changes and cannot rank multiple candidate solutions. Retrieval therefore provides the flexibility and scalability necessary for enterprise support environments.

**Clustering Historical Cases.**   For each KB article $k \in \mathcal{K}$, we collect its associated case summaries $\{s_1, \ldots, s_m\}$ and encode them as $\{\mathbf{h}_1, \ldots, \mathbf{h}_m\} \subset \mathbb{R}^d$. We use NV-Embed (Lee et al., 2024) as the encoder due to its strong semantic performance and optimized inference speed for the online phase. The overall pipeline is encoder-agnostic and supports substitution with any embedding model suited to the domain.

Hierarchical agglomerative clustering (HAC) with cosine distance and average linkage is applied to form coherent sub-problem clusters. We retain only leaf- level clusters and discard those below a minimum size to filter out noise and unstable groupings.[2] HAC is preferred over k-means as it does not require fixing the number of clusters in advance, allowing each KB to flexibly represent heterogeneous issue variants.

**KB-wise vs. Global Clustering.**   We cluster cases on a per-KB basis rather than globally. Global clustering risks merging cases from different KBs, diluting supervision signals and producing ambiguous anchors. KB-wise clustering preserves case–KB associations and lets each KB represent a variable number of sub-problems.

**Cluster Representations.**   Each cluster $C_{k,j}$ is represented by its mean embedding:

$$\mathbf{c}_{k,j} = \frac{1}{|C_{k,j}|} \sum_{\mathbf{h}_i \in C_{k,j}} \mathbf{h}_i,$$

yielding a multi-centroid representation $\{\mathbf{c}_{k,1}, \ldots, \mathbf{c}_{k,M_k}\}$ for KB $k$. To ensure stable and semantically coherent centroids, clusters below a minimum size are filtered out prior to index construction.

**Index Scale.**   We constructed three indexes: (i) a case-text index built from raw descriptions, (ii) a summary index built from domain-adapted issue summaries, and (iii) a KB-content index without clustering. HAC produced 790 clusters for the case-text index and 462 clusters for the summary index. The summary index is more compact due to cleaner, less noisy summaries.

**Retrieval in the Asymmetric Configuration.**   At inference time, a raw case description $c$ is embedded into $\mathbf{h}_c$ and scored against the summary-based index:

$$\text{score}(c, k) = \max_{j=1, \ldots, M_k} \cos(\mathbf{h}_c, \mathbf{c}_{k,j}),$$

where cosine similarity defines the match strength. This asymmetric setup (raw-case query → summary index) is both accurate and efficient, avoiding query- time LLM summarization while benefiting from the semantically aligned clustered index.

---

[2]We set the minimum cluster size to 3, balancing noise reduction with preserving meaningful sub-problems; see Appendix B.2 for a detailed sensitivity analysis.

**Confidence-Adaptive Hybrid Inference.** To further improve ranking quality, FineKB incorporates a confidence-adaptive inference mechanism. For a query $q$, let $c_1$ denote the top-1 centroid, with similarity $\gamma(q, c_1)$. This similarity acts as an uncertainty estimate:

$$\pi(q) = \begin{cases} \text{Fast Path,} & \gamma(q, c_1) \geq \tau_{\text{conf}}, \\ \text{Hybrid Path,} & \gamma(q, c_1) < \tau_{\text{conf}}. \end{cases}$$

*Fast Path (high confidence).* When similarity is high, FineKB assumes the relevant KB lies near $c_1$. It retrieves the top-5 embedding candidates and reorders the remaining four using cluster reliability $R(c)$:

$$\text{Pred1} = c_1, \quad \text{Pred2–5} = \text{top four candidates, sorted by } R(c).$$

No LLM or keyword search is invoked.

*Hybrid Path (low confidence).* When similarity is low, the query is treated as ambiguous. FineKB expands the candidate set via:

1. Pred1: top-1 centroid candidate,

2. Pred2: an LLM-generated recommendation (via `gpt-oss-120B`),

3. Pred3: the top-1 KB-content search result if distinct,

4. Pred4–5: remaining embedding candidates sorted by reliability.

**Cluster Reliability.** To complement geometric similarity with a data-driven estimate of centroid trustworthiness, we compute a reliability score for each centroid using the validation split. Cluster reliability measures how often a centroid leads to a correct KB prediction when it appears as the top-1 nearest cluster:

$$R(c) = \frac{|\{ q' \in \mathcal{D}_{\text{val}} : \text{top1}(q') = c \ \wedge \ \text{pred}(q') = \text{correct} \}|}{|\{ q' \in \mathcal{D}_{\text{val}} : \text{top1}(q') = c \}|}.$$

This score reflects the empirical precision of centroid $c$ over historical validation queries. During inference, $R(c)$ is used to refine the ranking of the lower-confidence candidates (ranks 2–5) within the fast path, providing a lightweight, retrieval-only re-ranking signal that improves robustness without incurring additional model calls.

**Unified Behavior.** In high-confidence settings, FineKB behaves as a pure vector re-ranker, using reliability to improve the top ranks. In low-confidence settings, it activates selective LLM reasoning and lexical search to recover cases where geometric retrieval alone is insufficient. Hybrid inference is triggered in only a small fraction of queries but yields meaningful accuracy gains with minimal impact on throughput.

## 4 EXPERIMENTS

We evaluate FineKB along two axes: (i) the effectiveness of domain-adaptive LLM summarization, and (ii) the retrieval performance of the cluster-aware index. This section describes the setup, presents summarization and retrieval results, and provides an analysis of model efficiency and accuracy.

### 4.1 EXPERIMENTAL SETUP

**Dataset Construction.** We curate a balanced corpus of 15,000 enterprise support cases drawn from 15 issue categories (1,000 cases per category). Each case contains (i) a raw problem description written by support agents or customers, and (ii) one or more ground-truth KB articles selected during case resolution (covering 188 unique KBs). We apply an 80/10/10 train–validation–test split with stratified sampling to preserve category-level balance. All clustering, indexing, and model-training steps use only the training portion; validation and test sets are held out for tuning and final reporting.

**Data Availability and Privacy.** Due to confidentiality constraints, raw textual content of enterprise cases and KB articles cannot be released. To support reproducibility while preserving privacy, we release only vectorized representations: (i) NV-Embed (Lee et al., 2024) embeddings of domain-adapted case summaries, and (ii) NV-Embed embeddings of KB article content. The repository includes full indexing pipelines, retrieval code, and scripts for reproducing all experiments.[3]

**Retrieval Evaluation Protocol.** A retrieved KB is considered correct if it appears in the ground-truth set associated with the case; all linked KBs are treated as positives. We report Recall@3/5, Mean Reciprocal Rank (MRR), and nDCG@5. Dense retrieval experiments use FAISS IVF-Flat (Douze et al., 2025) for all indexing and search operations.

**Baseline Configurations.** BM25 baselines use BM25S (Lù, 2024). ColBERT baselines rely on the RAGatouille implementation of ColBERTv2 with PLAID acceleration (Clavié, 2024; Santhanam et al., 2022). For HyDE, we follow the standard hypothetical-document retrieval pipeline: a large LLM (GPT-5.1) generates a pseudo-document from the raw case text, the pseudo-document is encoded using NV-Embed, and retrieval is performed over the same FAISS KB-content index used in the dense baseline. HyDE therefore incurs significant query-time LLM latency, which is reflected in the reported QPS. For FineKB Hybrid, selective LLM reasoning is performed using the `gpt-oss-120B` model, invoked only for low-confidence queries identified by the confidence-gated routing mechanism.

**Index Construction.** For all summary-based retrieval settings in Table 2, the retrieval index is constructed exclusively from teacher-generated summaries (`LLaMA-3.3-70B-Instruct`). This index structure remains fixed across all experiments; only the query representation varies in the symmetric settings (teacher summary vs. 8B-QLoRA summary). The *FineKB (Asymmetric)* configuration therefore consists of querying this high-quality teacher-summary index using the raw case text.

**Latency Reporting.** Reported QPS values reflect FAISS vector-search time only. In a production setting, encoding the raw case text using NV-Embed introduces an additional 30–50 ms of inference latency per query (measured on an NVIDIA A100-80GB MIG slice). FAISS search contributes less than 1 ms, resulting in an end-to-end retrieval latency of approximately 31–51 ms per case, which remains well within the acceptable bounds for real-time enterprise support workflows.

**Summarization Implementation.** We fine-tune `LLaMA-3.1-8B` on pseudo-labeled summaries using two strategies. For SFT, we train for 2 epochs (LR $2 \times 10^{-5}$, batch size 1, grad acc 32) using AdamW with gradient checkpointing. For QLoRA, we train for 2 epochs (LR $1 \times 10^{-4}$, batch size 4, grad acc 8) with rank $r = 32$, $\alpha = 32$, and dropout 0.05 targeting projection and feed-forward layers. QLoRA utilizes 4-bit NF4 quantization and bfloat16 compute precision.

**Summarization Metrics.** Quality is assessed using the FineSurE (Song et al., 2024) framework with GPT-5.1 as the evaluator. Ground-truth atomic facts are extracted from source texts using a teacher model (`LLaMA-3.3-70B-Instruct`). The evaluator judges if facts are retained, contradicted, or omitted to compute four metrics: **Factuality Error** (rate of hallucinations/unsupported claims), **Faithfulness** (fraction of strictly entailed statements), **Completeness** (coverage of ground-truth facts), and **Conciseness** (information density).

**Compute Environment.** LLM fine-tuning experiments were conducted on a system equipped with dual Intel Xeon Gold 6438M CPUs, four NVIDIA L40S GPUs, and 512 GB RAM (Python 3.10.12, PyTorch 2.6.0). Retrieval experiments ran on an A100-80GB GPU (2g.20GB MIG slice) via JupyterHub with up to 12 vCPUs. All latency and QPS measurements reflect direct on-hardware execution.

### 4.2 DOMAIN-ADAPTIVE SUMMARIZATION RESULTS

We compare the performance of full supervised fine-tuning (SFT) and quantized low-rank adaptation (QLoRA) against baseline instruction-tuned models. Table 1 presents the evaluation results across the four FineSurE dimensions.

---

[3]Embeddings and code: `https://github.com/ai-research-0084/finekb`

Table 1: FineSurE evaluation results (GPT-5.1 evaluator). `LLaMA-3.3-70B` serves as the reference upper bound. Among student models, `8B-SFT` achieves the lowest Factuality Error, while `8B-QLoRA` provides the best efficiency–quality tradeoff.

| Model | Factuality Error ↓ | Faithfulness ↑ | Completeness ↑ | Conciseness ↑ |
|---|---|---|---|---|
| `70B-Instruct` (Ref) | 29.1 | 72.4 | 99.3 | 99.5 |
| `8B-SFT` | **31.7** | 68.1 | 84.3 | 86.3 |
| `8B-QLoRA` | 33.1 | **68.2** | 83.1 | **89.2** |
| `8B-Instruct-SFT` | 36.2 | 64.6 | **89.9** | 45.7 |
| `8B-Instruct-QLoRA` | 36.3 | 63.9 | 89.0 | 50.4 |
| `8B-Instruct` (Base) | 53.4 | 49.2 | 83.3 | 87.0 |

Table 2: Retrieval and hybrid-inference performance across baselines and FineKB variants. Hybrid results incorporate confidence-gated LLM reasoning (4.1% activation). QPS values denote end-to-end throughput, combining FAISS lookup time with the expected cost of selective LLM invocation.

| Method | Index Construction | Query Representation | Retrieval / Inference Accuracy | | | | Efficiency |
|---|---|---|---|---|---|---|---|
| | | | R@3 | R@5 | MRR | nDCG@5 | QPS |
| *Baselines (KB Content)* | | | | | | | |
| ColBERT | KB Articles | Case Text | 20.71 | 26.56 | 0.15 | 17.88 | 134 |
| BM25 | KB Articles | Case Text | 29.33 | 40.21 | 0.23 | 27.26 | 151 |
| Dense (NV-Embed) | KB Articles | Case Text | 42.73 | 54.10 | 0.32 | 37.44 | **7,504** |
| HyDE (GPT-5.1) | KB Articles | Hypothetical Doc | 44.03 | 52.88 | 0.33 | 37.97 | 0.6 |
| *Baselines (Case Text)* | | | | | | | |
| BM25 | Case Text | Case Text | 60.28 | 70.59 | 0.48 | 53.42 | 138 |
| Dense (NV-Embed) | Case Text | Case Text | 63.85 | 74.74 | 0.50 | 56.54 | 7,185 |
| Dense (NV-Embed) | Summary (Single Centroid) | Case Text | 60.11 | 71.49 | 0.46 | 52.10 | 7,006 |
| FineKB (Symmetric) | Summary Clusters | Summary (70B-Instruct) | 59.22 | 71.41 | 0.49 | 53.07 | 2,833 |
| FineKB (Symmetric) | Summary Clusters | Summary (8B-QLoRA) | 59.95 | 71.00 | 0.49 | 52.66 | 2,935 |
| FineKB (Asymmetric) | Summary Clusters | Case Text | 65.39 | 77.34 | **0.55** | 59.08 | 3,030 |
| FineKB Hybrid | Summary Clusters + KB Articles | Case Text + LLM (4.1%) | **67.26** | **77.50** | 0.54 | **59.84** | 24 |

The zero-shot `8B-Instruct` baseline exhibits a high Factuality Error of 53.4%, confirming that instruction-tuned chat models are unreliable for technical summarization without domain adaptation. The best-performing adapted model, `8B-SFT`, reduces Factuality Error to 31.7%, closing the gap to the 70B teacher (29.1%) to within 2.6 percentage points. This demonstrates that the factual reasoning and fact-selection behavior of the teacher model can be effectively distilled into an 8B architecture.

Instruction-tuned variants (`Instruct-SFT/QLoRA`) show inflated Completeness scores (∼90%) but severely degraded Conciseness (∼45–50%) and increased hallucination rates, suggesting that chat-oriented priors conflict with the terse, high-precision style required for support incident summaries. Conversely, `8B-QLoRA` achieves competitive factuality (33.1%) and the highest Conciseness (89.2%).

In terms of training efficiency, full SFT required approximately 12 hours and 20 GB of checkpoints, whereas QLoRA completed in under 1 hour and produced only 327 MB of adapter weights. Given the marginal performance difference between SFT and QLoRA, the latter represents the most practical approach for scalable enterprise deployment.

## 4.3 CLUSTER-AWARE RETRIEVAL RESULTS

Retrieval was evaluated across three index types: (i) KB-article content, (ii) raw case text, and (iii) summary-derived cluster centroids (Table 2). Summary-based cluster indexes consistently achieve the strongest accuracy, particularly when queried with raw case descriptions.

The FineKB (Asymmetric) configuration—using LLM-normalized summaries for indexing and raw case text at query time—achieves **65.39% Recall@3** and **77.34% Recall@5**, outperforming all retrieval-only baselines while maintaining high throughput (3,030 QPS).

HyDE (GPT-5.1) provides a competitive generative baseline (44–53% recall), but its dependence on per-query LLM generation reduces throughput to only **0.6 QPS**. This confirms that separating the representation space (summary clusters) from the query space (raw narratives) is highly effective in this domain.

To evaluate end-to-end behavior, we also include the FineKB Hybrid mechanism (Section 3.3). The hybrid path was triggered for only **4.1%** of test cases, showing that the top-1 similarity score is a reliable confidence signal. The observed gains arise from three complementary components operating on low-confidence queries: (i) a reliability-based re-ranker that promotes historically precise centroids, (ii) a dense KB-content retrieval fallback that adds lexical grounding, and (iii) a single LLM-generated KB recommendation. Together, these yield **67.26% Recall@3** and **77.50% Recall@5**, improving over all retrieval-only configurations with minimal impact on throughput (24 QPS).

Symmetric variants of FineKB—where both the index and query use LLM-generated summaries—perform slightly below the asymmetric variant but still surpass all KB-content baselines. The parameter-efficient `8B-QLoRA` summarizer matches the 70B teacher within one point, demonstrating the practicality of lightweight domain adaptation.

Ablations confirm the necessity of multi-centroid modeling. Collapsing each KB into a single centroid yields only **60.11% Recall@3**, indicating that heterogeneous issue variants require structural representation. Dense retrieval over raw case text is competitive (**63.85% Recall@3**) but still below the asymmetric configuration due to narrative noise and redundancy. KB-content indexes perform worst overall (20–43% recall), underscoring the semantic gap between verbose case narratives and concise solution documents.

Overall, the results show that summary-based, cluster-aware indexing substantially improves retrieval quality, and that the confidence-gated FineKB Hybrid inference mechanism provides additional robustness for ambiguous cases while preserving enterprise-grade throughput.

## 5 CONCLUSION

This paper introduced FineKB, a domain-adaptive issue–summarization and cluster-aware retrieval framework that bridges the semantic gap between noisy enterprise case descriptions and curated KB articles. FineKB combines LLM-based issue summarization with per-KB multi-centroid indexing, transforming noisy case narratives into representations that align with the fine-grained structure of solution-oriented documentation.

The asymmetric configuration achieves 65.39% Recall@3 and 77.34% Recall@5, outperforming all dense and sparse retrieval baselines while maintaining high runtime efficiency. Although generative expansion via HyDE provides a competitive baseline, its dependence on per-query LLM generation leads to substantially lower throughput, making it impractical for real-time enterprise use.

Beyond retrieval, the FineKB Hybrid inference mechanism enhances end-to-end accuracy by combining three complementary signals: a reliability-based re-ranker, a KB-content dense-search fallback, and selective LLM reasoning for low-confidence cases. Despite activating the hybrid path for only 4.1% of queries, the system achieves 67.26% Recall@3 while maintaining an expected throughput of 24 QPS. These results highlight that targeted, confidence-gated reasoning can meaningfully improve retrieval quality without compromising scalability.

To support future research, we release the FineKB-Vectors dataset. Future directions include integrating FineKB with retrieval-augmented generation, extending retrieval to multilingual settings, and incorporating multimodal signals (e.g., logs, telemetry, screenshots) to improve robustness in hardware-centric support scenarios. Overall, FineKB provides a scalable framework for modern enterprise search, balancing the accuracy of generative techniques with the strict latency requirements of production systems.

## LIMITATIONS

While FineKB demonstrates strong improvements in retrieval accuracy, it has several limitations. First, although we release vectorized representations to enable statistical reproduction, the raw text of enterprise cases remains confidential. This prevents external researchers from performing qualitative error analysis or inspecting specific failure modes in the source narratives.

Second, pseudo-summaries generated by LLMs, while generally effective, may omit infrequent but critical technical details. Specifically, our evaluation shows the fine-tuned models achieve approximately 15–17% lower fact coverage (Completeness) than the teacher model, potentially impacting retrieval performance in rare edge cases where the omitted fact is the sole differentiator for a solution.

Third, FineKB relies on historical case evidence to construct multi-centroid representations. For newly created or rarely used KB articles with few or no linked cases, stable clusters cannot be formed. In such cold-start scenarios, the system naturally falls back to the KB-content dense index (NV-Embed), which remains fully supported in our pipeline but does not benefit from case-driven normalization. Developing lightweight summary generation or document-expansion strategies for cold-start KBs is an important direction for future work.

## ETHICS STATEMENT

In accordance with ICLR 2026 policy, we disclose that large language models (LLMs) were used solely to aid with writing polish, including grammar, style, and formatting. All scientific contributions, experiments, analyses, and claims were conceived and authored entirely by the human authors.

The experimental dataset consists of confidential enterprise case data. To balance scientific reproducibility with data privacy, we release only pre-computed vector embeddings of the dataset. Raw text processing and embedding generation were conducted in a secure environment following strict organizational governance, with all Personally Identifiable Information (PII) redacted prior to analysis.

## REPRODUCIBILITY STATEMENT

We provide detailed descriptions of the FineKB methodology in Section 3 and the experimental setup in Section 4. To facilitate reproduction of the retrieval results without compromising privacy, we release the **FineKB-Vectors** dataset, which includes:

1. NV-Embed embeddings of the domain-adapted case summaries (training/test sets), and

2. NV-Embed embeddings of the target KB articles.

These vectors allow for the exact replication of the cluster-aware retrieval experiments and metrics reported in Table 2. The anonymous repository additionally contains the full indexing pipeline, retrieval scripts, and hyper-parameter configurations used for the reported experiments.

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

# A    Issue Type Taxonomy

This appendix lists the primary issue categories used in FineKB. The taxonomy groups enterprise support cases into semantically coherent types that align with KB organization and downstream retrieval.

| Issue Type | Description / Notes | Sample Case Text |
|---|---|---|
| BATTERY | CMOS or controller battery warnings and failures | "System logs warning: controller battery capacity below threshold." |
| CONTRACT | Support or service contract-related issues | "Customer unable to open case due to expired service agreement." |
| CPU | Processor errors, thermal issues, or initialization failures | "CPU1 failed to initialize, system halts during POST." |
| DISK | Physical disk SMART errors, predictive failures, or offline drives | "Physical disk 0:1:3 failed, SMART predictive failure reported." |
| FAN | Cooling fan failures, abnormal speeds, or thermal alerts | "Fan3 operating below expected RPM, system at risk of overheating." |
| GPU | Graphics Processing Unit related failures or errors | "Server reports GPU PCIe training failure, device not detected at boot." |
| INFO | Customer inquiries or requests for product/feature information | "Customer asks how to check warranty status through the support portal." |
| MEMORY | DIMM/DRAM errors, ECC faults, or mismatched modules | "Uncorrectable ECC error on DIMM slot A2 during POST." |
| NETWORK | Connectivity, NIC, or switch configuration issues | "No link detected on embedded NIC port after reboot." |
| NO POWER | System fails to power on, no POST, or PSU not engaging | "Server does not respond to power button, only amber LED lit." |
| POWER SUPPLY | PSU-related faults, amber LEDs, or redundancy failures | "PSU2 failure detected, power redundancy lost." |
| RAID | Logical/physical array issues such as degraded state or rebuild failures | "Virtual disk degraded, one member drive missing from RAID5." |
| REMOTE ACCESS | Out-of-band or remote management controller issues | "Unable to connect to remote management interface, timeout error." |
| SOFTWARE | Operating system, firmware, or application-related problems | "Operating system fails to load after latest patch installation." |
| STORAGE CONTROLLER | Controller-related hardware issues (configuration, firmware, or battery warnings) | "Storage controller reports foreign configuration, unable to import array." |

Table 3: FineKB Issue Types with descriptions and representative sample case texts.

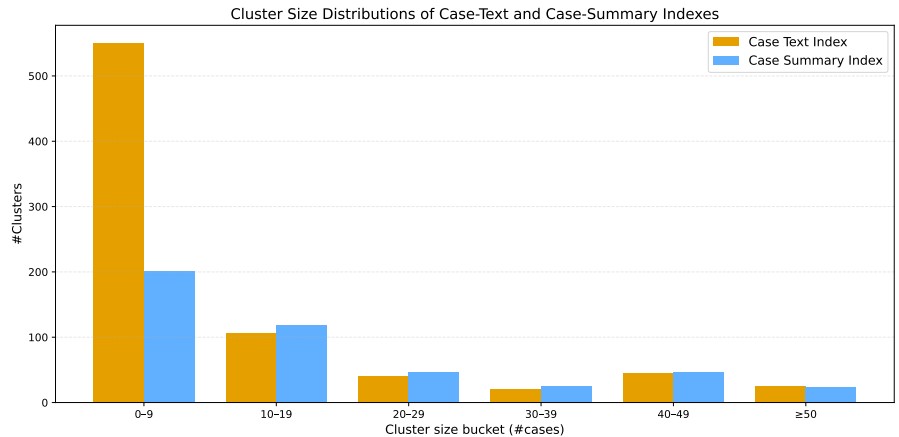

Figure 2: Distribution of cluster sizes (cases per centroid). The **Case Text** index (orange) is heavily skewed toward micro-clusters (0–9 cases), reflecting high semantic fragmentation. The **Case Summary** index (blue) yields a more balanced distribution with fewer, larger clusters, demonstrating improved semantic density.

## B  CLUSTER INDEX ANALYSIS

### B.1  INDEX TOPOLOGY

We report detailed statistics of the generated retrieval indexes in Table 4 and visualize their size distributions in Figure 2. The results highlight a key advantage of domain-adaptive issue summarization: the semantic consistency of the summaries allows the clustering algorithm to represent the knowledge space with significantly fewer, denser centroids.

Table 4: Index topology statistics. **Total Clusters** indicates the total number of centroids across the entire index. **Clusters per KB** ($k$) indicates the distribution of centroids allocated to represent each unique KB article (min/max/mean $k$).

| Index Source | # KBs | Total Clusters | Clusters per KB ($k$) | | | | |
|---|---|---|---|---|---|---|---|
| | | | Mean | Median | Min | Max | P75 |
| Case Text | 188 | 790 | 4.20 | 2 | 1 | 38 | 4 |
| Case Summary | 188 | 462 | 2.46 | 1 | 1 | 24 | 2 |

As shown in Table 4, the **Case Text** index requires a higher number of centroids (Mean $k = 4.2$) to cover the noisy, high-variance distribution of raw user narratives. In contrast, the **Case Summary** index achieves coverage with nearly half the number of centroids (Mean $k = 2.5$).

Figure 2 further illustrates this consolidation effect. The raw case-text index is dominated by micro-clusters (buckets of 0–9 cases), indicating that noisy text often fails to group meaningfully, necessitating many isolated centroids. Conversely, the summary index shifts the distribution toward larger buckets, confirming that summarization promotes better semantic aggregation and reduces index fragmentation.

### B.2  SENSITIVITY ANALYSIS: MINIMUM CLUSTER SIZE

To address the trade-off between index granularity and retrieval efficiency, we performed a sensitivity analysis on the *Minimum Cluster Size* hyperparameter using the FineKB (Asymmetric) configuration. This parameter controls the density of the index: smaller sizes retain more outliers, while larger sizes consolidate the index for efficiency.

As illustrated in Figure 3, the system exhibits high robustness for cluster sizes between 1 and 5. The analysis reveals a "sweet spot" at **Size 3** (Recall@3: 65.39%), which we adopted for the main

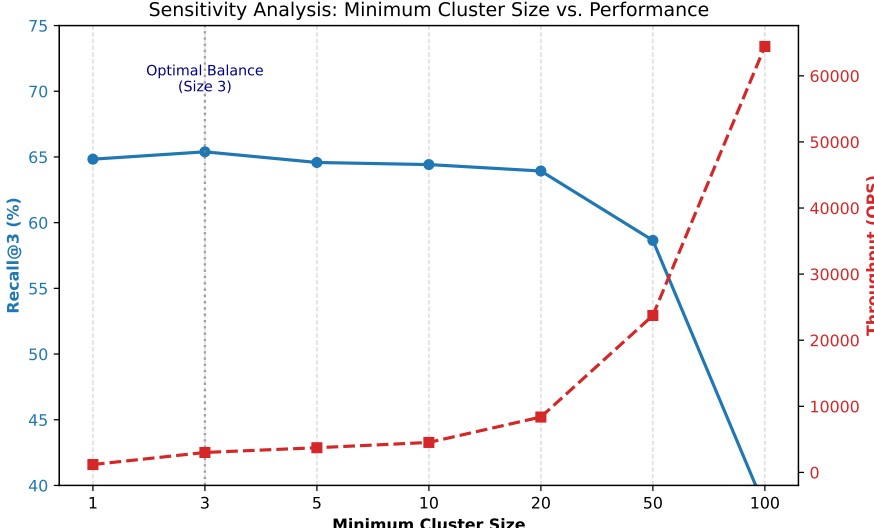

Figure 3: **Sensitivity Analysis of Minimum Cluster Size.** The blue line (left axis) shows that Recall@3 remains stable and effectively optimal at **Size 3** (65.39%), confirming that filtering micro-clusters removes noise without losing accuracy. The red dashed line (right axis) demonstrates that increasing minimum cluster size offers massive throughput gains (Pareto frontier), with only a moderate drop in recall.

experiments. Notably, removing singleton clusters (Size 1, Recall@3: 64.83%) actually *improves* performance slightly, confirming that very small clusters in this domain often represent noise rather than critical rare signals. Furthermore, the system exhibits a clear Pareto frontier where increasing the minimum cluster size to 10 improves throughput by nearly 50% (to 4,537 QPS) with only a marginal drop in recall ($\sim$1%), allowing for tunable deployment based on latency constraints.

