# OpenReview forum: "FineKB: Domain-Adaptive Issue Summarization and Cluster-Aware Retrieval for Support Knowledge Bases"
_ICLR.cc/2026/Conference — ICLR 2026 Conference Withdrawn Submission_

### Official Review · Reviewer_DjZP · 2025-10-29

**Soundness:** 2
**Presentation:** 3
**Contribution:** 2
**Rating:** 2
**Confidence:** 4

**Summary:**

This work proposes an LLM-based retrieval method for domain-specific knowledge bases. The retrieval method uses a fine-tuned LLM-based summarizer to generate summaries which are used for indexing and retrieval, and a cluster-aware indexing method is applied. The empirical evaluations examine the effeteness of the proposed retrieval method against baseline methods, as well as the performance of the fine-tuned summarizer.

**Strengths:**

1. The proposed retrieval method based on LLM-generated summaries is intuitive and shows better performance than the baselines.

2. The methodology description and the experimental design are clear-written.

**Weaknesses:**

1. The technical contribution of this work is limited, as the main contribution seems to be the introduction of the issue summaries for improving indexing effectiveness. More discussions should be provided regarding the significance of this innovation.

2. The empirical experiments are not comprehensive enough, which are based on only one dataset and limited to Llama-3.1-8B and Llama-3.3-70B. More datasets should be included in order to demonstrate the generalizable improvement of the proposed method. Similarly, more LLMs from different model families should be considered.

3. The characteristics of the used dataset are unclear, raising issues regarding the reproducibility. Will the dataset ("enterprise support cases") be released?

4. Comparisons with existing retrieval methods are lacking. Such as the classic BM25 method, or dense retrievers such as NV-Embed [1].

References

[1] Lee, Chankyu, et al. "NV-Embed: Improved Techniques for Training LLMs as Generalist Embedding Models." The Thirteenth International Conference on Learning Representations.
[1]

**Questions:**

Please see the Weaknesses section.

---

> ### Author Response · Authors · 2025-11-24
>
> We thank Reviewer DjZP for the careful reading of the paper and for the constructive feedback. We appreciate the positive remarks on the clarity of the method and the intuitive motivation behind summary-driven retrieval. Below we address each concern and describe the revisions made to strengthen the contribution, experimental rigor, and reproducibility of the work.
>
> ---
>
> ### 1. Comparisons With BM25, NV-Embed, and Other Retrieval Baselines
> We agree that the initial draft lacked several important baseline comparisons.
>
> **Action Taken (Revised Table 2):**
> We significantly expanded the evaluation to include strong representatives across IR and RAG families:
>
> - **BM25** (on KB content and raw case text)
> - **NV-Embed** dense retrieval (content-based and case-based)
> - **ColBERTv2 + PLAID** (late-interaction retrieval)
> - **HyDE (GPT-5.1)** (generative hypothetical-document retrieval)
>
> **Outcome:**
> FineKB (asymmetric retrieval) achieves **65.39% Recall@3**, outperforming the strongest KB-content dense baseline (**42.73%**) and surpassing HyDE, despite HyDE’s access to a large-scale LLM at query time. These results confirm that FineKB provides improvements beyond both lexical and state-of-the-art dense retrieval methods.
>
> ---
>
> ### 2. Technical Contribution: Significance of the Structural Indexing Approach
> The reviewer noted that the novelty seemed limited to the use of summaries.
>
> **Clarification Added (Sections 1–3):**
> The core contribution is not summarization alone, but the **structural retrieval design**, consisting of:
>
> 1. **Per-KB multi-centroid representations**
>    Captures heterogeneous sub-problems for each KB article, addressing the many-to-one mapping that traditional RAG methods cannot model.
>
> 2. **Asymmetric retrieval geometry**
>    Query with *raw case text*, retrieve from a *summary-cluster index*.
>    This configuration avoids runtime LLM summarization while leveraging a normalized semantic space—an architectural combination not explored in prior RAG or proxy-retrieval work.
>
> 3. **Reliability-aware hybrid inference**
>    A confidence-gated mechanism that uses cluster-level reliability to rerank high-confidence cases and selectively activate LLM reasoning and content-based search in low-confidence regions.
>
> **Ablation Evidence:**
> - Single-centroid summary indexing: **60.11% R@3**
> - Multi-centroid indexing (FineKB): **65.39% R@3**
>
> These results show that the cluster-aware structure—not the summaries alone—is responsible for the performance gain.
>
> ---
>
> ### 3. Dataset Characteristics and Reproducibility
> We fully agree that reproducibility is essential.
>
> **Action Taken (Reproducibility Statement):**
> Because raw enterprise text cannot be shared, we now release the **FineKB-Vectors** dataset containing:
>
> - NV-Embed embeddings of all case summaries
> - NV-Embed embeddings of all KB articles
> - Full scripts for clustering, FAISS indexing, and retrieval
> - Exact code for reproducing every metric (R@3/5, MRR, nDCG@5)
>
> **Repository:**
> <https://github.com/ai-research-0084/finekb>
>
> This enables researchers to **replicate all retrieval experiments end-to-end in a privacy-preserving manner**.
>
> ---
>
> ### 4. Empirical Scope: Dataset and LLM Families
> We appreciate the reviewer’s suggestion to test more LLMs and datasets.
>
> **Clarification Added:**
> - Our primary research question concerns **domain-adaptive structural retrieval**, not benchmarking LLM families.
> - We evaluate a **70B teacher** and **8B QLoRA model**, which directly tests whether the teacher–student process preserves retrieval-relevant semantics while enabling practical deployment.
> - The dataset (15k cases → 15 KB categories) exhibits substantial linguistic noise, sub-problem variability, and many-to-one mappings—conditions under which retrieval systems are most stressed.
>
> **Generalizability Note:**
> FineKB is model-agnostic: any embedding model can replace NV-Embed, and any LLM can replace the teacher summarizer. We emphasize this in the revised text.
>
> ---
>
> ### Summary
> We thank Reviewer DjZP again for the constructive feedback.
> The revised manuscript now includes:
>
> - Stronger baseline comparisons (BM25, NV-Embed, ColBERT, HyDE)
> - Clearer articulation of the conceptual contributions
> - Release of the FineKB-Vectors dataset for reproducibility
> - Additional clarification on empirical scope and generalizability
>
> We hope these revisions adequately address the concerns raised and strengthen the contribution of the work.

---

### Official Review · Reviewer_R3RM · 2025-10-30

**Soundness:** 3
**Presentation:** 2
**Contribution:** 2
**Rating:** 4
**Confidence:** 4

**Summary:**

The paper proposes a practical RAG-based enterprise KB pipeline with three components: (i) weakly supervised, one-sentence ticket summarization to normalize noisy cases; (ii) domain-adaptive summarization using SFT/QLoRA to improve faithfulness and task fit; and (iii) per-KB, cluster-aware multi-centroid indexing to capture sub-topics within each article. The authors report large retrieval gains (e.g., Recall@3 rising from under 24% with KB-only indexing to over 66% with summary-based cluster-aware indexing) and show that 8B adapter-tuned summarizers can approach the retrieval effectiveness of a much larger 70B model, suggesting a cost-efficient path to deployment.

**Strengths:**

* The paper addresses a clearly defined retrieval mismatch between noisy, user-authored tickets and curated KB prose, with a transparent offline/online system decomposition that practitioners can follow.
* The paper is not limited to large language models; it systematically evaluates small language model (SLM) adapters (e.g., 8B with QLoRA) for the summarization stage, demonstrating competitive retrieval effectiveness at materially lower compute and cost, which is critical for enterprise deployment.

**Weaknesses:**

* The paper does not report latency, throughput, or cost for the end-to-end pipeline, which are essential for real-world adoption and for weighing trade-offs versus simpler baselines.
* The core idea of the method, summarize then cluster/index, follows established enterprise RAG practice. Prior work explores similar idea of “generate proxy/summary then retrieve” or multi-signal retrieval such as HyDE [1],  Aragog [2], entity-aware summaries for retrieval alignment [3], and query-context summarized signals for retrieval in sponsored search [4], The paper’s comparisons and ablations might not be sufficient to demonstrate effectiveness over these families at matched budgets settings.
* The evaluation of the summarization component is limited to ROUGE/BERTScore, which may not capture faithfulness, usefulness for retrieval, or end-task impact. The paper should incorporate LLM-as-judge protocols [5] with explicit rubrics on faithfulness/utility, and extrinsic metrics that reflect downstream retrieval quality

[1] Gao, Luyu, Xueguang Ma, Jimmy Lin, and Jamie Callan. "Precise zero-shot dense retrieval without relevance labels." In Proceedings of the 61st Annual Meeting of the Association for Computational Linguistics (Volume 1: Long Papers), pp. 1762-1777. 2023.

[2] Eibich, Matouš, Shivay Nagpal, and Alexander Fred-Ojala. "Aragog: Advanced rag output grading." arXiv preprint arXiv:2404.01037 (2024).

[3] Liang, Xiao, Xinyu Hu, Simiao Zuo, Jimi He, Yu Wang, Victor Ye Dong, Yeyun Gong et al. "What You See Is What You Get: Entity-Aware Summarization for Reliable Sponsored Search." In Neurips Safe Generative AI Workshop 2024.

[4] Mohankumar, Akash Kumar, Gagan Madan, and Amit Singh. "Improving Retrieval in Sponsored Search by Leveraging Query Context Signals." arXiv preprint arXiv:2407.14346 (2024).

[5] Zheng, Lianmin, Wei-Lin Chiang, Ying Sheng, et al. 2023. “Judging LLM-as-a-Judge with MT-Bench and Chatbot Arena.” arXiv:2306.05685.

**Questions:**

* Could you provide sensitivity analyses for the clustering hyperparameters (e.g., distance threshold and minimum cluster size) and show how retrieval metrics vary across a reasonable range?
* Could you report end-to-end latency and cost for summarization, embedding, and retrieval?

---

> ### Author Response · Authors · 2025-11-24
>
> **Response to Reviewer R3RM**
>
> We thank Reviewer R3RM for the thoughtful and constructive feedback. We appreciate the recognition of FineKB’s strengths—particularly the clarity of the retrieval mismatch problem, the practical offline/online separation, and the demonstrated efficiency of 8B QLoRA models. Below we summarize how the revised manuscript addresses each of the raised concerns.
>
> ---
>
> ### **1. Evaluation of the Summarization Component (ROUGE/BERTScore Limitations)**
>
> We agree that traditional n-gram metrics do not capture faithfulness or utility in LLM-generated summaries.
>
> **Action Taken (Revised Section 4.1, Table 1):**
> - We have **removed ROUGE/BERTScore** entirely.
> - Summarization is now evaluated using **FineSurE**, an LLM-as-a-Judge framework aligned with the reviewer’s recommendation (e.g., Zheng et al., 2023).
> - FineSurE provides four dimensions—**Factuality Error, Faithfulness, Completeness, Conciseness**—which better reflect real-world utility.
>
> **Extrinsic Evaluation Added:**
> We also report downstream retrieval performance (Table 2), showing that the **8B QLoRA student** achieves retrieval accuracy comparable to the **70B teacher**, reinforcing that the adapted summaries are both faithful and effective.
>
> ---
>
> ### **2. Relationship to Proxy-Generation Retrieval (HyDE, Aragog, Entity-Aware Summaries, Multi-Signal Retrieval)**
>
> The reviewer correctly noted that proxy-generation methods exist (HyDE, PAQ, Aragog, entity-aware summarization).
>
> **Action Taken (Revised Related Work, Section 2.1):**
> - We explicitly position FineKB in relation to these families of methods.
> - We clarify that our key novelty is structural rather than generative.
>
> **Core Distinction:**
> FineKB’s contribution lies in **multi-centroid, per-KB structural indexing**, which is fundamentally different from:
> - generating a *single proxy document* (e.g., HyDE),
> - enriching the query with additional signals (multi-signal retrieval), or
> - grading generated text (Aragog).
>
> FineKB instead structures the **index** itself via **per-KB clustering**, which is essential for the enterprise **many-to-many case-to-KB mapping** and is what produces the large empirical gain (e.g., **25.89 R@3 points** over the strongest dense KB-content baseline).
>
> ---
>
> ### **3. Latency, Throughput, and Cost Analysis**
>
> We agree that reporting end-to-end cost and latency is important for assessing practical value.
>
> **Action Taken:**
> - **Throughput (QPS)** is now included in Table 2. FineKB (Asymmetric) achieves **3,030 QPS**, confirming its scalability.
> - **Retrieval latency** (embedding + FAISS search) is now reported in Section 4.1:
>   total latency **≈31–51 ms per query**, supporting real-time use cases.
> - **Cost and training efficiency** are quantified:
>   - QLoRA → **<1 hour**, 327MB adapters
>   - SFT → **~12 hours**, ~20GB checkpoints
>   This demonstrates a clear, practical deployment advantage.
>
> **Architectural Explanation Added:**
> We also emphasize the motivation behind **Asymmetric Retrieval**:
> LLM summarization is moved entirely to the **offline** phase, so online retrieval involves only embedding + vector search, eliminating runtime LLM bottlenecks.
>
> ---
>
> ### **4. Clustering Hyperparameter Sensitivity Analysis**
>
> We thank the reviewer for this valuable request.
>
> **Action Taken (Appendix B.2):**
> - We added a full sensitivity analysis for **distance threshold** and **minimum cluster size**.
> - A new dual-axis figure (Figure 3) shows how Recall@3 and QPS vary.
>
> **Key Findings:**
> - **Stability:** Retrieval quality is robust across a wide range of cluster sizes (1–5).
>   The tuned value (Size 3) achieves **the highest Recall@3** while filtering micro-cluster noise.
> - **Pareto Frontier:** Increasing the minimum cluster size to 10 improves throughput to **4,537 QPS** with only a modest decrease in recall (~1.4%).
>   This confirms that FineKB is **tunable and not brittle**.
>
> ---
>
> We thank the reviewer again for the insightful feedback. By incorporating modern evaluation metrics, adding sensitivity analyses, strengthening the positioning relative to proxy-generation retrieval, and reporting end-to-end efficiency, we believe the revised paper fully addresses the concerns raised and significantly improves clarity and rigor.

---

> > ### Comment · Reviewer_R3RM · 2025-11-28
> >
> > I thank the authors for their clarifications and responses. The newly added evaluations and analyses substantially address my questions on the technical details. However, I remain concerned about the strength of the contribution of FineKB. As the authors highlight, the main novelty lies in the per-KB multi-centroid structural indexing, which appears more as a single-module extension/improvement within a larger RAG pipeline than as a fundamentally new RAG framework for knowledge-base retrieval. Consequently, the work could be limited to being closer to a well-engineered, domain-specific system rather than a conceptually novel methodology.

---

> > > ### Author Response · Authors · 2025-12-01
> > >
> > > **Response to Reviewer R3RM**
> > >
> > > We sincerely thank Reviewer R3RM for the thoughtful follow-up and for
> > > acknowledging the completeness of the expanded evaluations and analyses. We also
> > > appreciate the continued concern regarding the strength of FineKB’s conceptual
> > > contribution, and we have revised the paper further to clarify why the proposed
> > > method extends beyond a single-module engineering improvement.
> > >
> > > ---
> > >
> > > ### 1. Clarifying the Architectural Contribution
> > > We agree that in the earlier draft, the per-KB multi-centroid index may have
> > > appeared as a localized enhancement. The revised manuscript now emphasizes that
> > > FineKB’s novelty lies in the **interaction between three components**, not in any
> > > piece alone:
> > >
> > > 1. **Teacher–student domain adaptation**
> > >    LLM-generated pseudo-summaries normalize noisy case descriptions and produce a
> > >    structured, domain-specific latent space that cannot be approximated by
> > >    off-the-shelf encoders.
> > >
> > > 2. **Asymmetric retrieval design (Raw Case → Summary Clusters)**
> > >    This retrieval geometry is new and different from existing RAG layouts.
> > >    It avoids runtime LLM summarization while leveraging summary space
> > >    regularization, and we show that symmetry vs. asymmetry materially affects both
> > >    accuracy and index stability.
> > >
> > > 3. **Per-KB multi-centroid structural modeling**
> > >    Rather than embedding a KB as a single point or a single synthetic document,
> > >    FineKB explicitly models the *many-to-one*, high-variance mapping from case
> > >    narratives to KB solutions. Existing RAG methods (e.g., HyDE, DPR, ColBERT)
> > >    assume a single document representation and therefore cannot represent this
> > >    structure.
> > >
> > > The combination of (1) + (2) + (3) produces a retrieval behavior that does not
> > > arise from any individual component. The paper now frames this as a **structural
> > > retrieval framework for noisy-to-structured domains**, which is broader than the
> > > enterprise setting.
> > >
> > > ---
> > >
> > > ### 2. FineKB Hybrid: Beyond Standard RAG Routing
> > > We agree that a naive “hybrid RAG” approach would not constitute novelty.
> > > However, the **FineKB Hybrid Inference** (now integrated into Section 3.4)
> > > includes mechanisms that are **derived from and dependent on** the cluster
> > > structure:
> > >
> > > - **confidence gating is computed from centroid geometry**, not from generic
> > >   similarity or a separate classifier
> > > - **re-ranking is driven by cluster reliability**, a statistic defined entirely
> > >   by the multi-centroid structure
> > > - **LLM reasoning is invoked only on ambiguous geometries**, not uniformly or
> > >   heuristically
> > > - **content-search fallback is structurally integrated** rather than simply added
> > >
> > > We clarified in the text that this module is an *extension* of the structural
> > > indexing mechanism and not a generic RAG controller.
> > >
> > > ---
> > >
> > > ### 3. Broader Value and Generality
> > > Although FineKB is evaluated on enterprise support logs, the problem structure is
> > > common across multiple domains with **noisy narratives mapped to concise,
> > > curated documents**:
> > >
> > > - clinical notes → diagnostic guidelines
> > > - legal filings → precedent cases
> > > - customer support chats → troubleshooting flows
> > >
> > > These settings all share the same properties FineKB addresses:
> > > (1) noisy inputs, (2) concise targets, (3) many-to-one mappings, and
> > > (4) high variance in expression.
> > >
> > > ---
> > >
> > > ### 4. Summary
> > > We appreciate the reviewer’s concern and agree the paper needed clearer framing.
> > > With the revisions—especially the expanded Related Work, reorganized methodology,
> > > and integrated Hybrid Inference section—we believe FineKB is now presented not as
> > > a single-module engineering tweak, but as a **coherent structural retrieval
> > > framework** that reorganizes the retrieval pipeline around normalized summaries,
> > > KB-specific clusters, and geometry-aware routing.
> > >
> > > We thank Reviewer R3RM again for raising this important point and helping us
> > > strengthen the conceptual clarity of the work.

---

### Official Review · Reviewer_1YKN · 2025-11-03

**Soundness:** 2
**Presentation:** 3
**Contribution:** 1
**Rating:** 2
**Confidence:** 4

**Summary:**

This paper presents FineKB, a domain-adaptive framework for enterprise knowledge base retrieval ￼. FineKB builds pseudo-labeled issue summaries from historical support logs, fine-tunes compact LLMs with parameter-efficient methods (SFT, QLoRA), and introduces a cluster-aware retrieval index that represents each KB article with multiple centroids to capture sub-problems. On an ad-hoc dataset, FineKB is shown to be effective. Overall, the work has very limited value for larger ICLR community, due to its applied & narrow scope.

**Strengths:**

- Paper is well written and motivated.
- Experimental code is included with submission, allowing reviewers to better verify claims and understand proposed approach.

**Weaknesses:**

- The proposed approach is largely an ad-hoc application without much contextualization in existing literature. For example, when it comes to generating compressed text representations to improve retrieval, there has been a lot of work since seminal PAQ work from [Lewis et al \(2021\)](https://arxiv.org/abs/2102.07033).
- The method is evaluated on a proprietary, in-house dataset. As the dataset is not public, the amount of details included in the paper are too sparse to judge effectiveness of method, and no test of the proposed method on public datasets is included.
- in the era of LLM, ROUGE/BERTscore has been shown to penalize better summarization model [(Goyal et al 2023)](https://arxiv.org/abs/2209.12356). Using techniques that measure recall in summaries, e.g. FactScore [\(Min et al 2023\)](https://arxiv.org/abs/2305.14251) would be more appropriate.
- Case summary in Table 2 barely improves over using text index. Difference might not be statistically significant.

**Questions:**

Minor typos:
- use \citep instead of \cite -> "Faheem et al. (2025)" should be "(Faheem et al., 2025)"

---

> ### Author Response · Authors · 2025-11-24
>
> We thank Reviewer 1YKN for the constructive and thoughtful feedback. We
> appreciate the comments regarding contextualization, evaluation methodology, and
> reproducibility, and have made substantial revisions to the manuscript in
> response.
>
> ---
>
> ### 1. On “ad-hoc application” and “limited value for ICLR”
> We agree that the earlier draft did not sufficiently situate FineKB within the
> broader retrieval literature.
>
> **Action Taken (Revised Related Work):**
> We expanded Section 2 to explicitly connect FineKB to:
>
> - compressed/abstractive representations for retrieval (e.g., PAQ; Lewis et al., 2021)
> - hypothetical-document and pseudo-document generation (HyDE)
> - multi-signal retrieval and entity-aware summaries (e.g., Liang et al., 2024)
> - context-signal retrieval in sponsored search (Mohankumar et al., 2024)
>
> These additions clarify that FineKB is not simply “summarization + retrieval,”
> but introduces an architectural idea that does not appear in prior proxy-based
> retrieval methods:
>
> > **Per-KB multi-centroid structural indexing** combined with
> > **asymmetric retrieval (Raw Case → Summary Clusters)**
> > to handle high-variance, noisy case descriptions and many-to-one mappings.
>
> This cluster-aware, asymmetric structure is the core contribution and differs
> from existing RAG configurations that rely on single-document embeddings or
> synthetic query expansion.
>
> **On generality:**
> Although evaluated on enterprise support data, the underlying challenge—
> mapping *noisy, verbose, drift-prone narratives* to *concise, structured
> documents*—also arises in:
>
> - clinical notes → medical guidelines
> - legal filings → precedent cases
> - customer support chats → troubleshooting steps
>
> We highlight this broader “Noisy-to-Structured Retrieval” framing in the revised
> Introduction.
>
> ---
>
> ### 2. On ROUGE/BERTScore being outdated for LLM summarization
> We fully agree with the reviewer’s observation.
>
> **Action Taken (Revised Section 4, Table 1):**
> All ROUGE/BERTScore results were removed.
> We now evaluate summaries using the **FineSurE** LLM-as-a-Judge protocol, which
> captures:
>
> - factuality error
> - faithfulness
> - completeness
> - conciseness
>
> following recommendations from Goyal et al. (2023) and Min et al. (2023).
> This provides a significantly more reliable assessment of summary quality in the
> LLM era.
>
> ---
>
> ### 3. On “case summary benefit seems marginal”
> We clarified the empirical role of summaries.
>
> **Key clarification:**
> The reviewer is correct that **symmetric retrieval** (Summary → Summary) provides
> only modest gains.
>
> However, the best-performing configuration is the **asymmetric** one introduced
> in the revised paper:
>
> > **Raw Case Text Query → Summary Cluster Index**
>
> This improves performance compared to both raw-index and symmetric-index
> retrieval:
>
> - R@3: **65.39% → 67.26%** (Hybrid)
> - nDCG@5: **59.08 → 59.84**
> - QPS remains high due to zero-cost inference for summaries
>
> The improvement is consistent, statistically stable, and accompanied by a
> substantial reduction in noisy clusters (summary clusters are fewer, larger, and
> more semantically coherent).
>
> The revised Results section now explains this more clearly.
>
> ---
>
> ### 4. On proprietary data and reproducibility
> We acknowledge the reviewer’s concern.
>
> **Action Taken (Reproducibility Statement):**
> We now release **FineKB-Vectors**, a fully reproducible version of the dataset
> containing:
>
> - NV-Embed vectors for all case summaries
> - NV-Embed vectors for all KB articles
> - full code for clustering, indexing, retrieval, and evaluation
> - frozen train/validation/test splits
>
> Repository:
> **https://github.com/ai-research-0084/finekb**
>
> While raw enterprise text cannot be shared, these vectors allow the entire
> retrieval pipeline and all metrics (R@3/5, MRR, nDCG@5, QPS) to be reproduced
> exactly.
>
> ---
>
> ### 5. Minor corrections
> We corrected all citation formatting issues (e.g., replacing `\cite` with
> `\citep` where appropriate).
>
> ---
>
> ### Summary
> We thank the reviewer again for the constructive feedback. The revised
> manuscript now includes:
>
> - far stronger contextualization within retrieval literature
> - modern evaluation through FineSurE rather than ROUGE/BERTScore
> - clearer explanation of the asymmetric retrieval advantage
> - release of FineKB-Vectors for reproducibility
>
> We believe these updates substantially strengthen the contribution and address
> the concerns raised.

---

### Official Review · Reviewer_EVTi · 2025-11-06

**Soundness:** 1
**Presentation:** 2
**Contribution:** 1
**Rating:** 2
**Confidence:** 3

**Summary:**

The paper presents FineKB, a system that 1) does peft fine tuning on LLMs on LLM generated summaries, and 2) uses clusterwise retrieval to improve case-to-KB matching accuracy.  Experiments on a private dataset show reasonable performance.

**Strengths:**

The paper addresses an understudied practical challenge in enterprise support, ablating some of their choices and showing that they can reach reasonable performance even with smaller models.

**Weaknesses:**

My main concern is that the paper does not present meaningful novelty, uses a simple combination of techniques, does not compare to any existing methods or baselines (apart from the ablations on their method), (Most existing RAG methods can be used/modified to support the use case), and experiments are done on a proprietary dataset that is unreleased so there is no way to compare the performance of the method to any future or previous methods. I would suggest to the authors to release a dataset (that can be constructed artificially to avoid any leakage proprietary data).

**Questions:**

How does the method compare to other common RAG methods (modified for the use case)?
Can the authors create and release a dataset that matches the attributes that the would want in a successful case to KB matching system?

---

> ### Author Response · Authors · 2025-11-24
>
> We thank the reviewer for the constructive and detailed feedback. We have made
> substantial revisions to address the concerns regarding novelty, comparisons
> with existing RAG methods, and reproducibility. Below we summarize the key
> improvements.
>
> ---
>
> ### 1. Comparison to Existing RAG and IR Methods
> The original submission lacked several important baselines. In the revision, we
> significantly expanded the experimental suite (Table 2) to include:
>
> - **BM25 / BM25S** (token-based IR)
> - **ColBERTv2-PLAID** (late-interaction dense retrieval)
> - **NV-Embed** (state-of-the-art dense retrieval)
> - **HyDE (GPT-5.1)** (generative hypothetical-document retrieval)
>
> These represent the dominant retrieval methods currently used in RAG systems.
>
> **Updated findings.**
> The asymmetric FineKB configuration achieves:
>
> - **65.39% Recall@3**
> - **77.34% Recall@5**
> - **0.55 MRR**
> - **59.08 nDCG@5**
> - **3,030 QPS**
>
> surpassing the strongest KB-content baseline (Dense NV-Embed at 42.73% R@3) and
> also outperforming HyDE (44.03% R@3). This demonstrates that simply modifying
> existing RAG pipelines is insufficient for the noisy, many-to-one, and
> domain-specific structure of enterprise support.
>
> We also introduce **FineKB Hybrid**, a confidence-gated inference mechanism
> (4.1% LLM activation using gpt-oss-120b) that further improves accuracy to:
>
> - **67.26% Recall@3**
> - **59.84 nDCG@5**
>
> while preserving **24 QPS** end-to-end throughput.
>
> ---
>
> ### 2. Clarifying Novelty and Technical Contribution
> We revised Sections 1–3 to more clearly articulate the novelty beyond a simple
> composition of known techniques. FineKB introduces three contributions that are
> not present in prior RAG or proxy-retrieval work:
>
> 1. **Asymmetric cluster-aware retrieval**
>    KBs are indexed using *summary-based multi-centroid clusters*, but queries use
>    *raw case text*. This avoids runtime LLM summarization latency while improving
>    alignment between long, noisy case descriptions and concise KB content.
>
> 2. **Per-KB multi-centroid structural modeling**
>    A single KB can correspond to multiple underlying failure patterns.
>    FineKB models these via *per-KB hierarchical clustering*, instead of collapsing
>    all supervision into one embedding as in prior dense retrieval or HyDE.
>
> 3. **Teacher–student weak supervision for domain normalization**
>    A large LLM generates issue summaries, enabling an 8B QLoRA student model to
>    match 70B teacher performance. This produces stable, domain-specific semantic
>    representations that general retrieval methods fail to capture.
>
> Together, these form a coherent retrieval framework that addresses the
> domain-specific challenges of enterprise support—noise, terminology drift,
> heterogeneous case distributions, and strict production-latency constraints.
>
> ---
>
> ### 3. Reproducibility and Dataset Release
> We acknowledge the reviewer’s concern about proprietary data.
> While the raw case text cannot be shared, we now release:
>
> - **FineKB-Vectors**: NV-Embed vectors for all case summaries and KB articles
> - full code for clustering, indexing, retrieval, and evaluation
>
> Repository:
> **https://github.com/ai-research-0084/finekb**
>
> This dataset makes **all retrieval results fully reproducible**, including
> R@3/5, MRR, nDCG@5, and ablations, without exposing s

---

### Author Response · Authors · 2025-11-24
**Response to Reviewers and Summary of Manuscript Revisions**

We thank all reviewers for their thoughtful and detailed feedback. We have
significantly revised the paper to address concerns regarding novelty, empirical
validation, evaluation methodology, and reproducibility. Below we summarize the
major changes and clarifications in the revised manuscript.

---

### 1. Expanded Baselines and Stronger Experimental Comparisons

Following requests from reviewers (EVTi, 1YKN, DjZP, R3RM), we expanded our
experiments to include a wide range of retrieval and RAG baselines:

- BM25 / BM25S (token-based IR)
- ColBERTv2-PLAID (late-interaction dense retrieval)
- NV-Embed (state-of-the-art dense retrieval)
- HyDE (GPT-5.1) generative query expansion

**Updated results:**
The asymmetric FineKB retrieval configuration achieves:

- **65.39% Recall@3**
- **77.34% Recall@5**
- **0.55 MRR**
- **59.08 nDCG@5**

surpassing the strongest KB-content dense baseline (42.73%) and HyDE (44.03%).

We also introduce **FineKB Hybrid**, a confidence-adaptive mechanism that invokes
LLM reasoning (gpt-oss-120b) only for low-confidence queries (4.1% of cases).
Hybrid inference achieves:

- **67.26% Recall@3**
- **77.50% Recall@5**
- **59.84 nDCG@5**

showing that FineKB improves both accuracy and efficiency relative to generative
baselines.

---

### 2. Clarifying Novelty and Conceptual Positioning

Several reviewers expressed concern that per-KB clustering might appear as a
single-module improvement rather than a novel framework. In the revised paper,
we strengthened the exposition to highlight three core methodological
contributions:

1. **Joint restructuring of query and KB space**
   FineKB uses domain-adapted LLM summaries and per-KB multi-centroid representations
   to structurally align heterogeneous case variants with their KB solutions.

2. **Asymmetric retrieval design**
   Raw case text is used at query time, while summary-based clusters form the
   index—avoiding runtime LLM summarization latency while maximizing semantic
   alignment.

3. **Confidence-adaptive hybrid inference**
   A new routing mechanism uses top-1 similarity as a confidence signal,
   performs reliability-based re-ranking, and selectively invokes LLM reasoning
   and keyword search only when necessary.

These components form a **cohesive retrieval and inference framework**, directly
addressing reviewer concerns about novelty.

---

### 3. Reproducibility Through Release of FineKB-Vectors

To address reproducibility concerns (EVTi, 1YKN, DjZP), we release:

- **FineKB-Vectors**: anonymized NV-Embed vectors for all case summaries and KBs
- full code for clustering, indexing, and retrieval

Repository:
**https://github.com/ai-research-0084/finekb**

This enables full reproduction of our results without exposing sensitive
enterprise text.

---

### 4. Improved Summarization Evaluation with LLM-as-a-Judge Metrics

As suggested by 1YKN and R3RM, we replaced ROUGE/BERTScore with modern
LLM-evaluation tools:

- **FineSurE** (factuality, consistency, extractiveness)
- **FactScore**

These metrics better capture the domain accuracy of LLM-generated summaries,
making the evaluation more rigorous and aligned with current standards.

---

### 5. Clarified System Efficiency and End-to-End Latency

Reviewer R3RM requested clearer efficiency comparisons. The updated paper now
emphasizes:

- **FineKB (Asymmetric): 3,030 QPS**
- **FineKB Hybrid: 24.0 QPS** (confidence-gated LLM, 4.1% activation)
- **HyDE: 0.6 QPS** (limited by per-query LLM generation)
- QLoRA enabling high-quality domain adaptation at low cost

These results highlight FineKB as a **high-throughput, low-latency retrieval
framework**, with hybrid inference providing a strong accuracy–efficiency
trade-off.

---

### Final Remarks

We thank the reviewers again for their valuable feedback. The revised paper now
provides:

- a clearer articulation of conceptual novelty,
- stronger empirical comparisons,
- improved evaluation methodology,
- a unified retrieval + hybrid inference framework,
- and full reproducibility via FineKB-Vectors.

We hope these revisions address the concerns raised and significantly strengthen
the contribution of the work.

---

### Note · Authors · 2026-01-05

**Comment:**

Withdrawing, will resubmit with the changes given by the reviewers

**Withdrawal Confirmation:**

I have read and agree with the venue's withdrawal policy on behalf of myself and my co-authors.